# Exploring the Role of Extracellular Vesicles in Skeletal Muscle Regeneration

**DOI:** 10.3390/ijms25115811

**Published:** 2024-05-27

**Authors:** Cristiana Porcu, Gabriella Dobrowolny, Bianca Maria Scicchitano

**Affiliations:** 1DAHFMO-Unità di Istologia ed Embriologia Medica, Sapienza Università di Roma, 00161 Roma, Italy; cristiana.porcu@uniroma1.it; 2Sezione di Istologia ed Embriologia, Dipartimento di Scienze della Vita e Sanità Pubblica, Università Cattolica del Sacro Cuore, 00168 Roma, Italy; 3Fondazione Policlinico Universitario A. Gemelli IRCCS, 00168 Roma, Italy

**Keywords:** extracellular vesicles, skeletal muscle damage, muscle regeneration, miRNAs

## Abstract

Skeletal muscle regeneration entails a multifaceted process marked by distinct phases, encompassing inflammation, regeneration, and remodeling. The coordination of these phases hinges upon precise intercellular communication orchestrated by diverse cell types and signaling molecules. Recent focus has turned towards extracellular vesicles (EVs), particularly small EVs, as pivotal mediators facilitating intercellular communication throughout muscle regeneration. Notably, injured muscle provokes the release of EVs originating from myofibers and various cell types, including mesenchymal stem cells, satellite cells, and immune cells such as M2 macrophages, which exhibit anti-inflammatory and promyogenic properties. EVs harbor a specific cargo comprising functional proteins, lipids, and nucleic acids, including microRNAs (miRNAs), which intricately regulate gene expression in target cells and activate downstream pathways crucial for skeletal muscle homeostasis and repair. Furthermore, EVs foster angiogenesis, muscle reinnervation, and extracellular matrix remodeling, thereby modulating the tissue microenvironment and promoting effective tissue regeneration. This review consolidates the current understanding on EVs released by cells and damaged tissues throughout various phases of muscle regeneration with a focus on EV cargo, providing new insights on potential therapeutic interventions to mitigate muscle-related pathologies.

## 1. Introduction

Skeletal muscle regeneration consists of a highly coordinated sequence of events that recapitulate the process of embryonic skeletal muscle development [1,2]. In the early stage of muscle regeneration, the injured muscle fibers undergo necrosis, and cellular contents, together with chemotactic factors, are released into the extracellular space, activating the inflammatory response [3]. The inflammatory phase is characterized by recruiting inflammatory cells, mainly neutrophils and macrophages, at the injury site. Inflammation represents a crucial step in the regenerative process, since it is responsible for muscle debris removal and the activation of satellite cells [4,5]. The next event is the regenerative phase, which is distinguished by the proliferation of activated satellite cells (SCs), also known as muscle stem cells [6]. Located between the sarcolemma and the basal lamina surrounding the myofibers, these cells typically remain quiescent under normal circumstances [7,8]. However, in response to various stimuli such as exercise, muscle growth, or injury, they become activated, enter the cell cycle to proliferate briefly, and subsequently differentiate into new myotubes or fuse with damaged myofibers for muscle repair. Following this activation and proliferation, SCs can return to a quiescent state, reconstituting the in vivo stem cell reservoir in preparation for subsequent regeneration processes [9,10,11]. The final phases of myofiber regeneration include the remodeling and maturation of muscle tissue, where reinnervation is established, the deposition of the extracellular matrix being initiated, and the recovery of the functional performance of the injured muscle [12,13].

It appears clear that efficient muscle regeneration is a coordinated process in which multiple factors are activated in sequence to restore and/or preserve muscle structure and function after an injury, and it is strictly dependent on fine communication between different cell types [5]. In this context, a key role is played by extracellular vesicles (EVs), which originate from the inward budding of the endosome membrane [14,15]. Extracellular vesicles are composed of lipid bilayers with diameters ranging from the inward budding of the endosome membrane. Extracellular vesicles are composed of lipid bilayers with diameters ranging from 20–30 nm [16] to 10 μm [17], secreted from almost all living cells, and detected in various body fluids and tissues [18,19,20,21,22]. EVs are released extracellularly with the functional purpose of transferring their informative cargo into targeted cells [23]. Macromolecules, such as lipids, proteins DNA, and RNA species like mRNAs, tRNAs, rRNAs, lncRNAs, and microRNAs, are enriched within EVs and exosomes, and their composition is related to the differentiation stage, environmental conditions, and epigenetic status of the parental cells [24].

MicroRNAs (miRNAs) are small non-coding RNAs approximately 19–24 nucleotides in length that function as negative regulators of gene expression. They exert their effects by binding to target messenger RNAs (mRNAs), leading to mRNA degradation or inhibition of protein translation [25,26,27,28]. A subset of miRNAs, called myomiRs, are exclusively or preferentially expressed in striated or enriched muscle, and are important factors in skeletal muscle myogenesis [29,30,31,32]. Indeed, it has been demonstrated that they participate in the regulation of muscle growth and differentiation, and ablation of miRNA results in muscle abnormalities and perinatal death [33,34]. MiRNAs can be secreted from cells in a stabilized form, bound either to proteins or within extracellular vesicles such as exosomes. MiRNA-associated exosomes play a role in cell-to-cell communication, which can either be communication between neighboring cells or distant cells in a different organ [35].

The involvement of extracellular vesicles, including exosomes, during skeletal muscle regeneration has been extensively studied in the literature, and here we aim to review the most recent publications regarding their involvement in different phases of skeletal muscle regeneration.

## 2. The Inflammatory Phase

Inflammation represents a crucial and necessary phase in muscle regeneration. Upon muscle injury, the damaged muscle fibers release chemotactic factors to the extracellular space, which induces the infiltration of many types of immune cells [3]. Macrophages represent the most abundant inflammatory cell population in the regenerative process, exerting both a proinflammatory and anti-inflammatory role. In fact, in the early phase post-injury, the proinflammatory macrophages, the M1 macrophages, remove cellular debris, stimulate satellite cell proliferation, and prevent premature myogenic cell fusion. M1 macrophages are followed by anti-inflammatory/promyogenic M2 macrophages, which support the differentiation of myoblasts and regulate the resolution of inflammation via the release of anti-inflammatory cytokines (Figure 1A) [36,37]. The macrophage’s crucial role in acute skeletal muscle injury repair is demonstrated by the fact that the absence of macrophage infiltration or disruption of macrophage functions leads to profound impairment of muscle regeneration and the development of muscle fibrosis [4,38,39,40,41].

Exosomes are secreted by a variety of cells, including macrophages [42], and recently it was demonstrated that miRNAs transferred via exosomes have a crucial role in regulating cellular functions [42]. Despite this, the extent to which macrophages communicate with muscle myoblasts via exosomes, especially through the transfer of exosomal miRNAs and exosomes’ role in muscle repair process, remains poorly understood. A recent paper by Zhou and colleagues demonstrated that in the C2C12 cell line, treatment with miR-501-enriched M2 exosomes promotes myoblast differentiation by targeting the ubiquitous transcription factor Yin Yang 1 (YY1) [43]. Additionally, through in vivo experiments, the authors demonstrated that M2 macrophage exosomes enhance inflammatory cell infiltration and exert a therapeutic effect on damaged pubococcygeal muscle in experimental models of stress urinary incontinence. These findings offer new insights into the promyogenic mechanism of M2 macrophages and suggest that M2 macrophage exosomal miR-501 could be a potential therapeutic tool to facilitate recovery from diseases resulting from muscle injury [43] (Table 1).

In addition, in a recent work published by Luo and colleagues, it was shown that under conditions of high post-injury inflammation, local myoblasts secrete miRNA-containing small EVs that can be taken up by surrounding macrophages, leading to incomplete M2 polarization and prolonged local inflammation, both in in vitro and in vivo experimental models. The authors demonstrated that miR-224, a miRNA differentially expressed in small EVs deriving from inflammatory myoblasts, could inhibit M2 polarization through a mechanism that likely involves direct suppression of its target gene. Furthermore, it has been demonstrated that miR-224 is upregulated by retinoblastoma (RB) protein phosphorylation and E2F1 release in myoblasts. The crucial role of miR-224 in the treatment of injured muscles was also demonstrated by the inhibition of miR-224 resulting in a dramatic improvement in muscle remodeling and functional recovery by promoting M2 polarization and alleviating inflammation [44] (Table 1).

## 3. The Regenerative Phase

The inflammatory response is followed by the regenerative phase (Figure 1B), marked by satellite cell activation and the appearance of regenerating fibers, which can be identified from a morphological point of view by the characteristic organization of nuclei in the central position and the expression of the embryonic/neonatal isoform of myosin heavy chain. When skeletal muscle undergoes damage, satellite cells (SCs) rapidly proliferate and subsequently fuse with the injured muscle, facilitating the formation of new muscle fibers [55]. This process plays a crucial role in promoting muscle growth, muscle remodeling, and overall repair after damage.

In this context, the injured plasma membrane of skeletal myoblasts and myotubes releases extracellular vesicles, including small EVs, which play a crucial role in mediating changes in target cells, and activating downstream pathways involved in muscle homeostasis and repair [56].

Exosomes, particularly those derived from mesenchymal stem cells (MSCs) or human skeletal muscle cells (HSkMs) can enhance regeneration in various tissues, including skeletal muscle [47,57,58]. Several studies show that MSC and HSkM exosomes enhance angiogenesis and myogenesis, resulting in increased muscle cross-sectional areas and reduced fibrotic tissue. These exosomes are rich in myogenic growth factors and miRNAs, promoting muscle cell proliferation and differentiation and preventing apoptosis. The genes involved in muscle regeneration, such as MyoD, myogenin, paired box 7 (Pax7), and embryonal myosin heavy chain (eMyhc), are upregulated in exosome-treated satellite cells or in exosome-treated injured muscle, leading to improved muscle function in vivo [59].

Notably, injecting exosomes derived from the C2C12 mouse myoblast cell line into the gastrocnemius muscle of injured mice promotes damage repair by triggering skeletal muscle regeneration and inducing the upregulation of Pax7 gene expression, a master regulator of SC function [60].

Muscle-derived exosomes carry specific lipids, such as cholesterol, sphingomyelin, ceramide, and lipid rafts, along with functional proteins such as contractile proteins (actins, myosins, and troponins) and myogenic growth factors [56,61]. Recently, Choi JS and colleagues demonstrated that exosomes released by differentiating human skeletal muscle cells carry myogenic growth factors, including heparin-binding EGF-like growth factors (HB-EGFs), vascular endothelial growth factors (VEGFs), insulin-like growth factors (IGFs), IGF-binding protein 3 (IGFBP-3), hepatocyte growth factors (HGFs), fibroblast growth factor 2 (FGF2), platelet-derived growth factors (PDGFs), interleukin 6 (IL6) and neurotrophin 3. These factors promote in vitro myogenic differentiation of stem cells and contribute in vivo to the regeneration of skeletal muscle tissue [61].

Furthermore, it is known that specific miRNAs, including miR-1, miR-133a, miR-133b, miR-206, miR-208a, miR-208b, miR-486, miR-499a, and miR-499b, can be detected in skeletal muscle [45], and their local injection triggered muscle regeneration in a rat model of skeletal muscle injury [62]. Recently, it has been demonstrated that four myomiRs, miR-1, miR-133a, miR-133b, and miR-206, are encapsulated in exosomes and are secreted by skeletal muscle tissues to facilitate communication between skeletal muscle tissues and neighboring cells during regeneration [63]. In addition, miR-206 and miR-1, loaded into satellite cell-derived exosomes, are also able to stimulate the differentiation of adipose-derived stem cells toward a myogenic lineage and can suppress collagen and fibronectin production by nearby fibroblasts, targeting the master regulator of collagen and fibrogenic expression, Rrbp1, finally promoting muscle regeneration [46] (Table 1).

## 4. The Remodeling Phase

The last stage of the regenerative process is the remodeling phase, in which cells talk and socialize amongst themselves to remodel muscle contractile structure, vascularization, and muscle–nerve communication, eliciting a full restoration of muscle functionality (Figure 1C). Several works have demonstrated that exosome administration in in vitro and in vivo models can strongly affect angiogenesis after muscle damage. Indeed, it has been pointed out that an abundance of specific miRNAs, including miR-126, miR-23a, and miR-494 [47,48], is crucial for the restoration of vessel integrity targeting angiogenesis-related pathways (Table 1).

Muscle damage can also involve peripheral nerve injury, and in this context, during the earliest stages of peripheral nerve regeneration, newly formed blood vessels contact Schwann cells and guide their migration into the area of injury, influencing their relationship with the regenerating axons [64]. Notably, the functional connection between muscle and motor neuron axons is restored by a communicative mechanism that involves extracellular vesicles [65,66]. In an elegant study, Madison and co-workers posited that EVs released by the injured neuromuscular junction can enter the nerve stump directly or influence axon regeneration via their distribution in the bloodstream [67].

The miRNA signature of exosome cargo during axon regeneration associated with muscle damage has not yet been characterized. However, in mouse models of neuropathy and nerve regeneration, 366 miRNAs, including those of the family of let-7, miR-125, miR-16, miR-103, miR-10a, miR-191a, miR-196, miR-21, miR-23a, miR-26a, miR-27b, miR-93, miR-99b, and miR-9a, have been identified as exosome cargo derived from Schwann cells. Interestingly, the highest-expressed miRNAs were linked to the regulation of axonogenesis, axon guidance, and axon extension, confirming the involvement of Schwann cell exosomes in axonal homeostasis. Moreover, it has been observed that circulating levels of myomiRNAs such as miR-206 and miR-133a during surgical or pathological denervation are modulated during the reinnervation process or the progression of disease, suggesting that muscles release messages to influence and restore the functional connection between muscles and nerves [49,50] (Table 1).

In addition to muscle angiogenesis and muscle reinnervation, small EVs contribute to extracellular matrix (ECM) remodeling of injured muscles. Indeed, small EVs derived from MSCs carry several immunomodulatory proteins, such as HGF, IL10, and angiogenic factors, including VEGF, and TGFβ that synergically orchestrate the functional restoration of the muscle tissue environment [68,69].

A crucial role in ECM remodeling during skeletal muscle regeneration is exerted by the fibroadipogenic cells (FAPs). FAPs are key players in muscle regeneration with multimodal action, since they can elicit the deposition of fibrotic tissue or can facilitate muscle regeneration based on the environmental niche where they are housed. In an interesting work, Sandonà and co-workers demonstrated that in dystrophic muscles, where muscle regeneration is impaired, FAP-derived EVs mediate functional interactions of FAP cells with muscle satellite cells, boosting muscle regeneration and reducing muscle fibrosis if enriched with miR-206, suggesting that the modulation of the exosome cargo can influence the restoration of muscle tissue structure and functionalities [51] (Table 1).

## 5. Role of Extracellular Vesicles in Skeletal Muscle Pathologies

It is known that skeletal muscle can respond to various external stimuli such as physical exercise, changes in hormonal balance, the availability of oxygen and nutrients, and the activity of motor neurons. However, the plasticity of skeletal muscle can be compromised under certain conditions, such as diabetes, disuse, denervation, and aging, resulting in disruption in the functional capabilities and regenerative potential of skeletal muscle. Indeed, most muscle pathologies are characterized by the progressive loss of muscle tissue due to chronic degeneration combined with the inability of regeneration machinery to replace the damaged muscle [70]. Sarcopenia, which is defined as the age-related loss of muscle mass and strength, represents one of the most significant challenges for older individuals since it is one of the main causes of impaired physical performance and reduced mobility compromising normal activities of daily living [71]. In recent years, several studies have highlighted the potential therapeutic role of exosomes in sarcopenia regulation. Indeed, as mentioned above, skeletal muscle differentiation is a highly regulated process that requires coordinated intercellular communication, in which exosomes play a crucial role. Exosomes released from skeletal muscles promote myoblast proliferation and differentiation and facilitate the transmission of vital signaling molecules between muscle cells, highlighting their key role in mediating communication between myoblasts and myotubes. Recent research conducted by Aswad et al. has shown that a high-fat diet administered to mice causes their skeletal muscles to release exosomes, stimulating myoblast proliferation and inducing alterations in gene expression associated with the muscle cell cycle and differentiation in vitro [72]. Furthermore, exosomes derived from other cell types, such as MSCs, increase the proliferation and differentiation of C2C12 cells, while adipocyte-derived exosomal miR-27a triggers insulin resistance in skeletal muscle cells [52] (Table 1).

In addition, it has been revealed that miR-29-enriched exosomes deriving from placental MSCs (PL-MSCs) increase the differentiation of human muscle cells. At the same time, the in vitro treatment with conditioned medium or exosomes secreted by PL-MSCs increased the differentiation of myoblasts and decreased the expression of fibrogenic genes in Duchenne muscular dystrophy (DMD) patient myoblasts. This evidence highlights the significant potential of exosomes as therapeutic agents in skeletal muscle sarcopenia and suggests that the targeted delivery of exosomal miR-29c may have important clinical applications in cell therapy of DMD [53].

Another important consequence of several pathological conditions and aging is the imbalance between protein synthesis and degradation that triggers muscle atrophy and severely compromises muscle function [73]. Atrophy is characterized by a decrease in muscle mass and at the histological level by a reduction in fiber cross-sectional area. It results in reduced force production, easy fatigue, and reduced exercise capacity, along with a lower quality of life [74]. It has been shown that among the main pathophysiological causes correlated with the onset of atrophy, there are high levels of oxidative stress, chronic inflammation, and decreased mitochondrial function, which then result in the stimulation of signaling pathways involved in the ubiquitin-dependent proteasome system, the lysosome and autophagy system, and mTOR activation [75,76,77,78]. One of the molecules that plays a crucial role in inducing skeletal muscle atrophy is IL6 [79]. In a recent paper, Xuan Su and colleagues revealed that stem cell-derived EVs with modification of miRNAs can attenuate muscle atrophy through an inhibitory effect on the IL6 pathway [80] (Table 1). Another study further demonstrated that atrophic muscle fiber-derived EV miR-690 inhibits satellite cell differentiation during aging-induced muscle atrophy. These findings provide new insight into sarcopenia and suggest a potential treatment strategy [54].

## 6. Extracellular Vesicle Delivery as a Therapeutic Tool for Tissue Repair and Regeneration

Extracellular vesicles, and in particular small EVs, exhibit accommodating features that suggest they would be promising therapeutic tools to promote the repair and regeneration of soft tissue, including skeletal muscle. Nanoparticles can produce therapeutic outcomes similar to cell therapies, with the added advantage of avoiding many drawbacks and rejections. Despite most of the experimental evidence supporting extracellular vesicle efficacy and safety, the studies performed have largely been confined to pre-clinical models. The development of exosomal treatments is still in its early stages, with limited clinical applications achieved so far. However, in this scenario, one significant advancement is the purified exosome product (PEP) developed by the Mayo Clinic, obtained from human plasma platelets. Currently, various applications of PEP, including tendon repair, peripheral nerve regeneration, vaginal tissue regeneration (phase I clinical trial; ClinicalTrials.gov: NCT04664738), myocardial infarction recovery (phase I clinical trial; ClinicalTrials.gov: NCT04327635) and amyotrophic lateral sclerosis ALS7 (NCT06249412) are undergoing phase I clinical trials [81] (Table 2).

Few studies have evaluated the safety and side effects of exosome-based treatments. Mesenchymal stem cells (MSCs) and plasma are two widely held sources for exosome use; however, the safety of their clinical translation faces several challenges. Indeed, it has been observed that plasma-derived EVs might carry molecules of diseased tissues and drugs with high toxic potential [82]. Conversely, tumor-derived EVs have been shown to deliver chemotherapeutics efficiently to recipient cells both in vitro and in vivo, but evidence regarding their safe use is still debated [83,84]. Therefore, to tackle safety concerns regarding exosome usage, it is recommended that a precise step-by-step approach [82] be enhanced. This approach should entail identifying a reliable and efficient source of therapeutic EVs, understanding the mechanisms underlying their therapeutic effects, and exploring potential modifications to enhance safety. Additionally, extracellular vesicles possess attributes aligning with an optimal drug delivery system, including a bilipid membrane structure, versatility in cargo transportation, and the capability to target tissues with minimal toxicity concerns. However, extracellular vesicle productivity is low and therefore represents another challenge for EV therapeutical use. To overcome this limitation and enhance EVs’ therapeutic potential, researchers have employed different strategies, including the optimization of exosomal cargo profiles by the overexpression of molecules related to therapeutic effects [46]. Indeed, it has been described that the enrichment of myotube-derived exosomes with miR-1, miR-133, miR-206, and miR-125b, miR-494, and miR-601 promotes a broad action of pro-regenerative cellular events such as macrophage polarization towards an anti-inflammatory and pro-regenerative state that, as mentioned above, are prodromic for an efficient and complete regenerative process [46]. Another methodological approach to compensate for EVs’ low productivity rate is an improvement in EV delivery to target cells. This can be performed by specific treatment of parental cells, exposing cells to exogenous compounds, or genetically manipulating parental cells to enhance exosome production. In addition, parental cell treatment can be used to tailor EVs to optimize their cargo profiles, thus improving their therapeutic effects [81].

A further strategy to ameliorate EVs’ therapeutic application is enhancement of EV biodistribution and tissue targeting. Indeed, intravenously injected exosomes are rapidly taken up by macrophages, which mainly accumulate in the liver and spleen, leading to a low number of extracellular vesicles delivered into the target organ. To overcome this limitation, ligand modification of extracellular vesicles has been proposed to attenuate EV accumulation in filtering organs and to enhance accumulation in target tissues [85]. This approach was particularly useful for the modulation of inflammation by targeting key cytokines, such as interleukin 6 (IL6). Indeed, in a recent work, EVs were engineered to carry an IL6 signal transducer to inhibit the IL6 intercellular signaling cascades and selectively attenuate the inflammatory response in muscle pathologies, such as DMD, without interfering with the anti-inflammatory IL6 pathway [86].

Finally, another challenge is the optimization of exosome storage conditions. Indeed, EV quality and integrity are crucial for targeting efficiency, cargo loading capacity, and EV stability. A commonly used method to preserve extracellular vesicles is storage at low freezing temperatures. It is widely demonstrated that the quantity and the quality of extracellular vesicles are well maintained at -80 C; however, the freezing process requires high shipping costs and special laboratory equipment, which represent a considerable complication for the wide use of EVs in clinics.

Another method that can guarantee EV stability at higher temperatures is lyophilization. The lyophilization (freeze-drying) technique is an established method for the isolation and storage of exosomes, and unlike the low-temperature method is a cost-effective and straightforward technique. However, little evidence has been collected to determine EV efficacy and shelf-life after lyophilization, and thus lyophilized EVs are far from being used in clinical application.

Therefore, despite the limitations described, extracellular vesicles are expected to become effective therapeutic agents for various diseases, although there are no standardized techniques for their isolation, purification, administration, or delivery.

Recently a new promising strategy has been reported that uses cellular nanoelectroporation to produce large quantities of small EVs loaded with therapeutic mRNAs and miRNAs. Unlike traditional methods that load therapeutics into EVs after their formation, this approach uses selected parental cells to generate therapeutic mRNAs from plasmid DNAs, incorporate them into EV precursors, and release them through exocytosis. This significantly improves the efficacy of EV therapies compared to those based on naturally released or externally loaded small RNAs [87].

In conclusion, the use of EVs in regenerative medicine is becoming increasingly attractive due to their high biocompatibility, low risk of immune response, and absence of cells. Despite their promise, EV therapies currently face challenges such as a limited RNA load, low production rates by parent cells, and a short lifespan within the body. Thus, establishing effective routes for EV injection and standard techniques for isolation, qualification, and purification becomes essential for the clinical translation of exosome use.

## 7. Conclusions

In the present review, we summarized the growing knowledge on the role played by extracellular vesicles in modulating the different phases of muscle regeneration. We elucidated that vesicle cargo can modulate in a specific manner the pathways involved in the inflammatory, regenerative, and remodeling phases, suggesting the possibility that exosomes can be used as a therapeutic strategy to influence skeletal muscle regeneration.

However, it must be pointed out that although the scientific interest in vesicle communication is increasing, exosome clinical translation is currently still limited. This is mainly due to the isolation and purification methodologies that hinder high-quality and large-scale exosome production, along with their long-term storage and in vivo stability. Therefore, further scientific efforts are necessary for the improvement of methodologies and to better elucidate exosome cargo to develop clinical tools for human diseases.

## Figures and Tables

**Figure 1 ijms-25-05811-f001:**
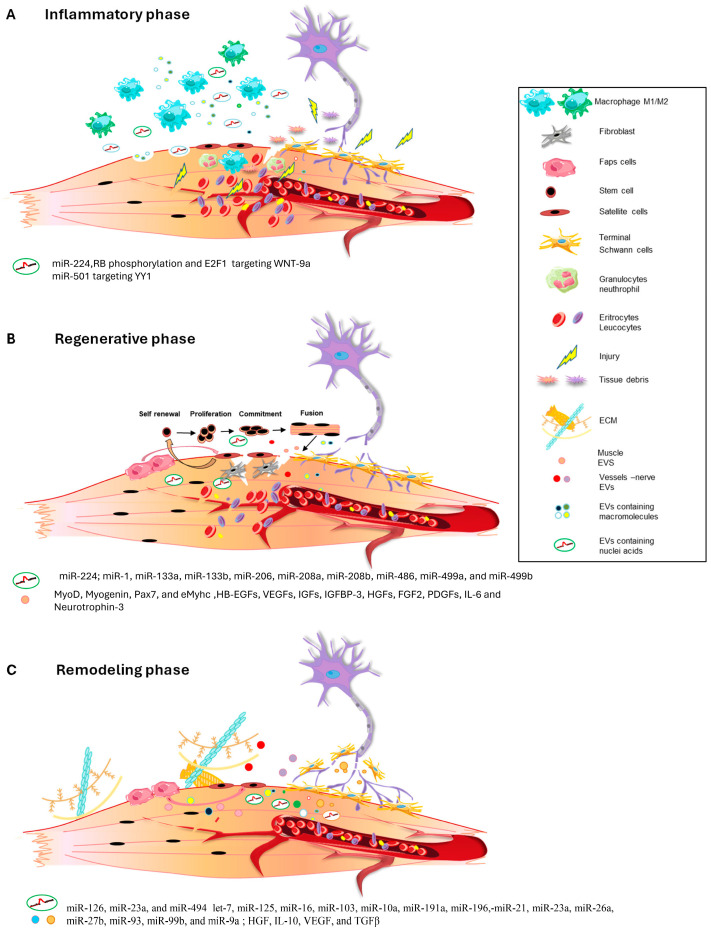
(**A**–**C**) Schematic representation of EVs release during the different phases of muscle regeneration process.

**Table 1 ijms-25-05811-t001:** Tissue-specific EV miRNAs associated with skeletal muscle disorders.

Tissue/Source EVs	miRNAs Cargo	Physiological Effect	Reference
M2 macrophages	miR-501	Decreased inflammation (Stress urinary incontinace/Pubococcygeal muscle)	[43]
Myogenic cells	miR224	Increased inflammation indamaged muscle	[44]
Myogenic cells	miR-1, miR-133a, miR-133b, miR-206, miR-208a, miR-208b, miR-486, miR-499a, and miR-499b	Improved regeneration of damaged muscle	[45]
Satellite cells	miR-1, miR-133a, miR-133b, miR-206	Muscle differentiation of adipose stem cells/inhibition of fibrotic tissue deposition	[46]
Mesenchymal stem cells	miR-126, miR-23a, miR-494	Restoration of vessels integrity in damaged muscle	[47,48]
Schwann cells	let-7, miR-125, miR-16, miR-103, miR-10a, miR-191a, miR-196,-miR-21, miR-23a, miR-26a, miR-27b, miR-93, miR-99b, and miR-9a	Axonogenesis/guidance in damaged muscle	[49,50]
FAPS cells	miR-206	Improvement of muscle regeneration in DMD	[51]
Adipocyte cells	miR27a	Insulin-resistance in Skeletal Muscle	[52]
Placental MSC	miR29c	Improvement muscle regeneration	[53]
Atrophic muscle fiber	miR690	Sarcopenia	[54]

**Table 2 ijms-25-05811-t002:** Current clinical trials related to EV intervention in skeletal muscle disorders.

Clinical Trial Identifier	Status	Disease or Condition	Phase	Intervention
NCT04664738	Active not recruiting	Skin graft, peripheral nerve regeneration, vaginal tissue regeneration	I	Purified Exosome Product
NCT04327635	Enrolling by invitation	Myocardial infarction recovery	I	Purified Exosome Product
NCT06249412	Not yet recruiting	Amyotrophic Lateral Sclerosis ALS7	Not applicable	Purified Exosome Product

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
