# Peer review of "Exploring the Role of Extracellular Vesicles in Skeletal Muscle Regeneration"

_ijms, 2024, doi:10.3390/ijms25115811_

Round 1

Reviewer 1 Report

Comments and Suggestions for Authors

The review of Porcu and colleagues provides a useful summary of recent studies on the role of Extracellular Vesicles (EVs) in skeletal muscle repair and homeostasis. Their functions during the different phases of muscle regeneration and on the regulation of myogenesis are extensively and clearly described and summarized in Figure 1.

In addition, the manuscript presents some insights regarding EVs in muscle diseases, as age-related sarcopenia and Duchenne muscular Dystrophy. Finally, authors describe the potential therapeutic interest of EVs, particularly of exosomes, as well as important safety concerns related to EV-based medical approaches.

As very minor corrections, to improve the quality of this review, I suggest to ameliorate the resolution of Fig1, which appears slightly blurred in the pdf document, and to change few definitions:

Line 99: add “by” before "in vivo experiments"

Line 141: C2C12 mouse myoblast “cell line” instead of cells

Line 170:  several “works” instead of “articles”

Line 216: “disruption” instead of “disruptions”

Line 238: “myoblasts” or “muscle cells” instead of “myoblast cells”.

As further suggestion, authors should include acronym definition when missing (e.g. rrbp1) and revise the manuscript to avoid some typos and repetitions.

Author Response

The authors are grateful to this Reviewer for giving them the opportunity to improve the manuscript. We hope we satisfied all your requests and suggestions which are highlighted in yellow throughout the text.

Please find below the answers to the comments of the Reviewers. 

Reviewer 1

  • As very minor corrections, to improve the quality of this review, I suggest to ameliorate the resolution of Fig1, which appears slightly blurred in the pdf document

Author answer:

Thank you for your suggestion. We improved the quality of Figure 1 in the revised version of the manuscript (600 dpi).

  • Line 99: add “by” before "in vivo experiments"

Author answer:

We satisfied this request

  • Line 141: C2C12 mouse myoblast “cell line” instead of cells

Author answer:

We satisfied this request

  • Line 170: several “works” instead of “articles”

Author answer:

We satisfied this request

  • Line 216: “disruption” instead of “disruptions”

Author answer:

We satisfied this request

  • Line 238: “myoblasts” or “muscle cells” instead of “myoblast cells”.

Author answer:

We satisfied this request

  • As further suggestion, authors should include acronym definition when missing (e.g. rrbp1) and revise the manuscript to avoid some typos and repetitions.
  • Author answer:

To better accomplish this request we introduced a list of Acronyms pag. 9-10.

Reviewer 2 Report

Comments and Suggestions for Authors

In the manuscript, the authors explore the current understanding of EVs released by cells and damaged tissues throughout various phases of muscle regeneration, with a focus on EV cargo, aiming to offer new insights on potential therapeutic interventions to mitigate muscle-related pathologies. The topic is interesting; however, the exploration of EVs as agents in regenerative medicine, particularly in preclinical and clinical settings, appears underdeveloped. I’d like to recommended a major revision.

1. Please reference MISEV2018 and MISEV2023 guidelines to accurately define small and large EVs, avoiding the generic term "exosome."

2. The authors are strongly recommended to list a table for current clinical trials related to skeletal muscle disorders.

3. The authors should consider summarizing the roles of microRNAs (miRNAs) associated with skeletal muscle disorders in a detailed table. This would offer a structured overview of how miRNAs influence these conditions.

4. While the focus on microRNAs in EVs is appreciated, the manuscript might benefit from expanding on engineered EVs that are enhanced with therapeutic proteins (doi.org/10.1016/j.biomaterials.2020.120435) and encapsulated mRNAs (doi.org/10.1002/advs.202302622) to address their relatively low therapeutic potency in their naïve state. This aspect could be thoroughly discussed in the discussion section to explore potential solutions.    

Author Response

The authors are grateful to this Reviewer for giving them the opportunity to significantly improve the manuscript. We hope we accomplished all your requests and suggestions which are highlighted in yellow throughout the text.

  • Please reference MISEV2018 and MISEV2023 guidelines to accurately define small and large EVs, avoiding the generic term "exosome."

Author answer:

We are grateful to the Reviewer for this suggestion. We inserted the reference related to MISEV2018 and MISEV2023 guidelines [14,15]. In addition, along the text, we changed the generic term exosomes with extracellular vesicles (EVs) or small EVs with the exception where the Author of the cited article referred specifically to exosomes (we highlighted the changes in yellow throughout the text.

  • The authors are strongly recommended to list a table for current clinical trials related to skeletal muscle disorders.

Author answer:

  • We introduced a table (Table 2) reporting the clinical trials related to the use of EVs in skeletal muscle disorders pag.7.

  • The authors should consider summarizing the roles of microRNAs (miRNAs) associated with skeletal muscle disorders summarizing the roles of microRNAs (miRNAs) associated with skeletal muscle disorders in a detailed table. This would offer a structured overview of how miRNAs influence these conditions.

Author answer:

We introduced a table (table 1) reporting the most important miRNAs associated with skeletal muscle disorders (pag.7).

  • While the focus on microRNAs in EVs is appreciated, the manuscript might benefit from expanding on engineered EVs that are enhanced with therapeutic proteins (doi.org/10.1016/j.biomaterials.2020.120435) and encapsulated mRNAs (doi.org/10.1002/advs.202302622) to address their relatively low therapeutic potency in their naïve state. This aspect could be thoroughly discussed in the discussion section to explore potential solutions.

Author answer:

We are grateful to the Reviewer for this suggestion. We introduced the references in the text and we discussed the therapeutic potential of engineered EVs. Pag 8 .Lines 327-332 and pag.9 lines 354-366 (highlighted in yellow).

Round 2

Reviewer 2 Report

Comments and Suggestions for Authors

The authors have satisfactorily addressed my concerns.

Comments on the Quality of English Language

Please pay attention to the typos in the manuscript. E.g., Table 1. "miRNas"; "decresed"; 

Author Response

The authors apologize for typos that have been corrected in Table 1 and throughout the text.